# Can EEG Correlates Predict Treatment Efficacy in Children with Overlapping ASD and SLI Symptoms: A Case Report

**DOI:** 10.3390/diagnostics12051110

**Published:** 2022-04-28

**Authors:** Slavica Maksimović, Ljiljana Jeličić, Maša Marisavljević, Saška Fatić, Aleksandar Gavrilović, Miško Subotić

**Affiliations:** 1Cognitive Neuroscience Department, Research and Development Institute “Life Activities Advancement Center”, 11000 Belgrade, Serbia; s.maksimovic@add-for-life.com (S.M.); m.marisavljevic@add-for-life.com (M.M.); s.fatic@add-for-life.com (S.F.); m.subotic@add-for-life.com (M.S.); 2Department of Speech, Language and Hearing Sciences, Institute for Experimental Phonetics and Speech Pathology, 11000 Belgrade, Serbia; 3Department of Neurology, Faculty of Medical Sciences, University of Kragujevac, 34000 Kragujevac, Serbia; agavrilovic@hotmail.rs; 4Department of Neurophysiology, Clinic of Neurology, University Clinical Centre of Kragujevac, 34000 Kragujevac, Serbia

**Keywords:** EEG correlates, auditory-verbal processing, ASD, SLI, integrative therapy, treatment efficacy

## Abstract

Evaluation of the rehabilitation efficacy may be an essential indicator of its further implementation and planning. The research aim is to examine whether the estimation of EEG correlates of auditory-verbal processing in a child with overlapping autism spectrum disorder (ASD) and specific language impairment (SLI) symptoms may be a predictor of the treatment efficacy in conditions when behavioral tests do not show improvement during the time course. The prospective case report reports follow-up results in a child aged 36 to 66 months. During continuous integrative therapy, autism risk index, cognitive, speech–language, sensory, and EEG correlates of auditory-verbal information processing are recorded in six test periods, and their mutual interrelation was analyzed. The obtained results show a high statistically significant correlation of all observed functions with EEG correlates related to the difference between the average mean values of theta rhythm in the left (F1, F3, F7) and right (F2, F4, F8) frontal region. The temporal dynamics of the examined processes point to the consistency of the evaluated functions increasing with time flow. These findings indicate that EEG correlates of auditory-verbal processing may be used to diagnose treatment efficacy in children with overlapping ASD and SLI.

## 1. Introduction

The paper presents a prospective, longitudinal study of a child with overlapping autism spectrum disorders (ASD) and specific language impairment (SLI) symptoms. In the two and a half years of continuous integrative therapy, behavioral tests and EEG findings were evaluated to estimate which variable best describes treatment progress.

Specific language impairment is diagnosed when a child’s language development is deficient in understanding and producing spoken language, which happens for no apparent reason [1]. The prevalence of SLI among children aged from 5 to 6 years was reported as 7.4% by Tomblin et al. [2], while the prevalence of autism spectrum disorder has been rapidly increasing in recent times [3] and ranges from 2.64% [4] to 2.93% [5]. Unlike SLI, ASD represents a spectrum of various manifestations, and according to the DSM-V, includes deficits in social communication and social interaction and restricted, repetitive behaviors, interests, and activities [6].

However, many studies and clinical observations have reported that language and communication deficits in children with SLI have also been present in children with ASD [7,8,9,10,11]. In a population of children diagnosed with ASD, about 63% have language impairment [12], while more than 50% have difficulties in different aspects of language, for example, phonology, grammar, and semantics [13]. Furthermore, brain imaging studies showed comorbidity of ASD and SLI through atypical hemispheric asymmetry in frontal regions [14]. Additionally, it should be noted that more than 60% of children with ASD receive treatment by speech and language pathologists [15].

Despite reports that litigate the possibility of comorbidity between ASD and SLI disorders [16,17], researchers have also documented strict distinctions between the two disorders [18,19,20]. The distinction between ASD and SLI is reflected in tasks with non-word repetition, which is significantly lower in SLI children [18,21]. Additionally, speech articulation is poorer in SLI children [17]. Comorbidity between ASD and SLI is reflected in poor vocabulary and expressive and receptive language deficits [18]. Pragmatic deficits are the most common symptom in the clinical picture of ASD [22].

Furthermore, several studies explored the distinctions between SLI and ASD using neuroimaging techniques, such as electroencephalography (EEG). Some of them detected neural biomarkers linked with patterns in the clinical picture of ASD [23]. EEG is a non-invasive method for measuring the electrical activity of brain regions [24]. Many studies investigated a large number of neural biomarkers that are in correlation with ASD symptoms, such as increased spectral power in the alpha activity of frontal regions in infants [25], left frontal theta [26,27], right frontal theta [28], and gamma spectral power [29]. Higher theta coherence in the frontal regions during a resting state [30] and in task conditions [31] was also observed. Other studies documented opposite results with decreased spectral power in theta [32] and alpha in the frontal and temporal regions [33]. Neural biomarkers that correlate with speech and language are linked with the temporal regions responsible for speech perception [34,35,36,37] and frontal regions bilaterally for speech production [38]. It was found that increased frontal theta during visual stimulation in infants is linked with verbal and non-verbal intelligence later during childhood [39,40]. Meyer et al. [41] found that increased frontal and temporal theta predicts better language and cognitive skills. Studies on infants documented the role of frontal theta in future cognitive abilities of children [39]. In studies with adults, theta is mostly linked with attention and working memory [42,43,44]. Kardos and colleagues proposed that higher frontal theta is a predictor of capacity in working memory in older participants [45]. In EEG studies in a resting state, delta and theta decreased, while alpha and beta increased with age [46,47,48]. Studies with typically developed children documented that theta waves are dominant in the anterior brain regions during speech perception in 0–3-year-old children [49] and continue up to 4 years old [47]. ASD symptoms typically manifest during early childhood, around two years [15]. Establishing an ASD diagnosis is a months-long and sometimes years-long process. During that process, ASD manifestations can evolve. The process is challenging due to the diagnostic procedure based on the child’s observation, taking data from the parent or guardian and medical records [50]. Extensive data are involved in diagnosing ASD [51]. Despite that, due to the complex nature of the disorder and insufficiently reliable genetic or biological diagnostic markers [52], and parents’ inadequate noticing of ASD symptoms in children [53], it is hard to determine a precise diagnosis.

A precise and early diagnosis of ASD can afford a proper education and treatment program [51]. According to Maw and Haga [54], early intervention of ASD relies on behavioral, cognitive, educational, and developmental approaches. Parents’ integration in stimulation therapy, especially for communication development, is essential [55,56]. Many studies gave an advantage for early intensive treatment because of the progressive impact on development in children with ASD [57,58,59,60], while one study reported the impact of early intervention in the long-term period [61].

Furthermore, the critical role of neuroplasticity and the critical period for brain development, from speech and language development, is well documented in the literature [62]. According to these findings, early treatment will reduce pathological complications of neurodevelopmental disorders through brain neuroplasticity and critical periods in which the developing brain is additionally more approachable than at other periods. According to one magnetic resonance (MR) study, a critical period for language acquisition in the brain starts from birth and may persevere until the age of seven [63]. Many authors recommend that the optimal time to start treatment is in 1–3 years of life [64,65]. The review study by Towle and colleagues [66] gave the example of twelve studies that support that early treatment in ASD children will provide later advantages in many children’s skills. Such findings impose the need for early intervention in children with ASD and SLI.

It is usually not easy to specify the final diagnosis during the early diagnostic process in young children (i.e., the first admission of a child). This is especially so if the child has not previously been involved in integrative therapy, but also due to the comorbidity between certain diagnoses [7,8,9,10,11]. Speech–language deficits are characteristics of ASD and SLI, while the existence of a common profile of them is a matter of tireless debate in the research community [7]. In that sense, the goal of this study was not to determine the final diagnosis at the time of the child’s admission but to monitor the child’s progress due to the implementation of early integrative therapy that ensures the development of the child’s existing abilities. In addition to the importance of implementing early treatment, an important aspect is the assessment of the treatment efficacy in a certain period. This aspect is essential due to the critical period of speech–language development [64,65,66].

There are many challenges in determining the efficacy of ASD interventions. Some of the mentioned challenges are highly variable symptoms in children with ASD, fluctuations in behaviors in short periods independent of any treatments, highly variable responses to intervention, and conducting multiple treatments simultaneously [67,68].

In general, assessing the efficacy of any intervention in children with ASD represents a crucial but demanding task. Accordingly, it is hard to predict which children will respond to particular treatments, what treatment intensity might make a difference, and what behaviors the therapy might affect [69].

Depending on the severity of the clinical picture of ASD and SLI, frequent slow progress in treatment leads therapists to be unsure whether the treatment is effective. In cases when behavioral tests do not show the child’s progress while the treatment lasts a long time, the question of the effectiveness and efficacy of the treatment arises.

To enhance our knowledge of the dynamics of therapy progress in children with ASD and SLI, we report the autism risk index and cognitive, speech–language, and sensory profile of a girl with overlapping ASD and SLI symptoms in a prospective, longitudinal study between 36 and 66 months of age. On the other hand, considering the possibilities of using electroencephalography as a non-invasive method in the assessment of linguistic and cognitive processing electrophysiological correlates, the question arises if such a method could indicate whether the applied treatment enhances the development of speech and language skills in children, while behavioral tests show no progress. This information would be of undoubted importance to therapists and could in some way contribute to treatment planning.

Accordingly, the study’s primary goal is to examine whether the estimation of EEG correlates of auditory-verbal processing has a diagnostic value and may predict treatment efficacy in conditions when behavioral tests do not register changes in the level of examined functions.

## 2. Materials and Methods

### 2.1. Case Report

The girl is the only child of young (mother 30 years, father 32 years), healthy, non-consanguineous parents. She lives in a monolingual family where the Serbian language is spoken. She has been enrolled in kindergarten since her second year of age. She was 36 months old when her parents brought her for an examination, and immediately after that, integrative therapy began. A multidisciplinary admission team examined the child at the institute for experimental phonetics and speech pathology (IEPSP) in Belgrade, Serbia. The team consisted of a psychologist, a speech–language pathologist, a psychiatrist, and a neurologist. The assessment was carried out first by the whole team and then by individual assessment of each team expert. In order to obtain anamnestic data, the assessment was carried out, starting with an interview with parents and inspection of medical records, followed by child observation and the application of certain diagnostic tests. The anamnestic data have been obtained from a detailed interview with the parents and the child’s medical documentation (the Labor and Delivery note, the assessment of general physicians, and otolaryngologists). She was born at 40 weeks of gestation after a normal pregnancy. Delivery was without complications. Apgar’s score was 9/10 and her birth weight was 3070 g. Pre- or perinatal risk factors were not present. Pre-lingual speech phases occurred regularly, but the lingual phase was inconsistent after the babbling phase. Early motor development was typical. Hearing screening tests indicated normal hearing. A hearing assessment performed using the brain-stem-evoked response audiometry (BERA) showed normal auditory function at the level of 20 db bilaterally. Specific family heredity did not exist.

At the time of admission, the child was non-verbal (without functional words), uncooperative, and manifesting ASD symptoms. During the assessment of the girl’s behavior, she showed a distinct level of excitement and irritability in the presence of strangers. She cried inconsolably, did not sit, walked from wall to wall of the room, and did not react to her parents’ attempt to calm her down. She did not accept any toys, not even manipulative ones. It was not possible to direct her to a specific activity or game. Her mother and father calmed her down with difficulty. She often pulled her mother’s hand to go outside. She expressed a tendency to avoid eye contact; she showed atypical responses to eye gaze. She did not respond to her name when called by her parents or others. According to the parents, she would react similarly in an unknown environment and in a familiar one, e.g., with grandparents who lived one floor below their apartment. Gradual “exclusion” from the social environment and behavioral specific manifestations began to manifest around 18 months. The problem deepened by going to kindergarten around the age of two. Generally observed, there was no interaction between the girl and the therapist during joint play with toys or some other activities. She did not initiate or share joint attention.

Compensatory use of non-verbal communication was missing—she did not use or understand demonstrative gestures (pointing, showing, and giving). She used her parent’s hand to achieve the desired goal. Social interaction with examiners was extremely poor. Furthermore, she had restricted behavior and play; she often visually explored toys. She manifested repetitive body movements like hand flapping and constantly moving around the room. Additionally, she had stereotypical behavior. The sensory profile manifested as disgust for specific textures and selecting some foods by color. Sleep was orderly. Diet was limited, with drinking large daily amounts of milk formula.

There was no adequate reaction and localization of low- or high-frequency sounds in space, even when they were of high intensity when she was in an activity. She showed reactions more strongly to particular auditory and visual effects from TV content (commercials, cartoons). Speech comprehension was markedly reduced. The girl barely understood the speech of her parents and the environment, and her understanding was at the level of a few simple experiential tasks that were situationally supported. Her reaction to a verbal order was faster and more reliable for simple, situational, verbal orders from the immediate social environment if they were related to her own need or current interest.

On the other hand, speech production was absent. The girl was non-verbal. She did not produce any functional words or holophrases during the first assessment. However, she used her central voice production and babbling without communicative function. Generally considered, the girl had severe receptive and expressive language deficits, and fell significantly behind the chronological age considering speech–language development. It was hard to distinguish whether the speech and language deficit manifested itself as an independent disorder or if it was within the ASD during the assessment at the child’s first admission.

Although the child was non-verbal at admission, it is important to emphasize that speech and language skills have developed with specific dynamics during the integrative therapy. However, it was estimated in later assessment periods when the child showed progress in speech–language development. In parallel with the development of speech and language, the girl’s autistic symptoms reduced as well.

Her motor development was typical (sitting at seven months, walking at 12 months). She used her right hand more often at admission, and was confirmed to be right-handed at an older age. Although she did not show specifics in motor skills development, the functional use of objects and toys was minimal; she took toys, looked at them briefly, and threw them without imitation of movement and play.

After observing and analyzing the obtained data, the multidisciplinary admission team made the so-called working diagnosis: ASD and overlapping SLI symptoms. It is essential to point out that it was hard to determine with certainty whether the child had ASD or SLI because she was 36 months old and had not previously had treatment. The admission team defined that only immediately started, continuous, and integral therapy can lead to progress in a child’s development. Considering all collected data about the child, the multidisciplinary admission team decided on a multidisciplinary implementation team to make an integrative therapy treatment plan. The team optimized the KSAFA approach [70] in the treatment according to the individual needs and specifics of the child. Applied integrative therapy included speech and language therapy, psychomotor re-education, sensory integration, and psychotherapeutic assistance to the child’s mother. More precisely, the multidisciplinary implementation team for all participants included:-A speech–language pathologist conducted speech–language therapy every day for 60 min during the first year, after which it was reduced to 2–3 three times per week.-A special educator conducted sensory integration therapy once a week for 60 min.-A special educator conducted psychomotor re-education two times per week for 60 min.-A psychologist conducted psychotherapy counseling sessions with the mother two times per month for 60 min.

Despite the very intensive treatment, the child showed no progress in examined developmental functions in the first six months of integrative therapy. She was inconsolable with the presence of crying and anger, refusing any contact with the tendency to widen that gap even more.

### 2.2. Study Design

At the starting time point (t0), and before the treatment, the initial assessment was performed: (1) cognitive assessment (The Čuturić Developmental Test (RTČ-P), REVISK); (2) Gilliam Autism Rating Scale (GARS-3); (3) Sensory profile 2; (4) Children’s Communication Checklist (CCC-2); (5) The scale for evaluation of psychophysiological abilities of children (SEPAC); (6) EEG recording in two conditions: (a) resting state and (b) during auditory-verbal stimulation.

Treatment started seven days after the initial evaluation. The evaluation was repeated every six months during the continuous treatment with the same instruments administered at point t0, except in the point t2, when the Children’s Communication Checklist was included in the additional observation—Second Edition CCC-2—examined by a psychologist, and from the point t4, when the psychologist administered another test for cognitive assessment, regarding the child’s age. More precisely, the dynamics of the child’s progress conditioned the application of mentioned tests at later ages.

Thus, evaluation was performed at the following points at ages: t0 = 36 months; t1 = 42 months; t2 = 48 months; t3 = 54 months; t4 = 60 months; t5 = 66 months.

The complete study protocol had been approved by the Ethics Committee of the Institute for experimental phonetics and speech pathology (Date: 15 February 2018, No 2/18-1) in Belgrade, Serbia, which operates in accordance with the ethical principles in medical research involving human subjects, established by the Declaration of Helsinki 2013. The child’s parents provided written informed consent to participate in this study.

### 2.3. Measures and EEG Recordings

(1) Cognitive profile was assessed at different times points with two different instruments depending on the child’s age:

The Čuturić Developmental Test (RTČ-P) [71] examines the psychomotor development of infants, young children and preschool children. It examines the development of psychomotor skills, oculomotor skills, emotionality, speech, auditory-motor reactions, communication and sociability, and verbal expression of knowledge. RTČ-P is created for examining children from 2 to 8 years of age and consists of seven subtests with six tasks each. Between subtests, there is continuity in tasks that involve handling individual objects. The child’s achieved result is expressed by the coefficient of mental development. Different materials (rattle, bell, pot, bottle and ball) are used in the test. The estimation was conducted in the presence of a parent during the first three assessment points (t0, t1, t2 and t3).

REVISK [72] is a revised Wechsler Intelligence Scale for Children in Serbian. According to Wechsler’s principle, the test is standardized for measuring children’s overall intellectual functioning and cognitive abilities. The instrument provides insight into total, verbal and performance scores, where higher scores reflect higher levels of intellectual functioning. The Verbal and Performance Scales consist of 5 subtests, where Information, Comprehension, Arithmetics, Similarities and Digit Span belong to the Verbal Scale subtests, and Picture Completion, Picture Arrangement, Block Design, Object Assembly and Coding belong to the Performance Scale subtests. The test was applied at t4 and t5 assessment points. At these two assessment points (60 and 66 months of age), the girl was verbal and had developed speech–language abilities.

(2) The Gilliam Autism Rating Scale, third edition (GARS-3) [73] is an instrument for ASD evaluation and consists of 58 statements, divided into six subscales: (1) restricted/repetitive behaviors; (2) social interaction; (3) social communication; (4) emotional responses; (5) cognitive style; (6) maladaptive speech. Each question from the subscale is graded from 0 to 3, where 0 means “does not apply to the child”, 1 means “partially applies to the child”, 2 “mainly refers to the child”, and 3 means “completely applies to the child”.

Autism Index (AI Score) is the sum of standard scores obtained by converting the raw scores of each subscale into a standard score. Two autism indexes may be computed. If the child is non-verbal, only four subscales are assigned, and autism index is based on those four subscales. On the other hand, all six subscales should be assigned if the child is verbal, so the autism index is based on all six subscales. AI score represents the risk of autism and the severity of the disorder. A score of ≤54 indicates that autism is unlikely to be present, and a score of 55–70 indicates that it is likely to be present (with severity level 1). Index 71–100 indicates a very probable presence of autism (with level 2), and index ≥ 101 indicates a very probable presence of autism (with level 3), which indicates severe ASD symptoms and requires a high level of support.

This test was applied in all assessments points as it assesses the severity of ASD symptoms, which may be essential in monitoring the clinical picture of a child with ASD who is included or not in the therapy. It must be noted that the subscales “cognitive style“ and “maladaptive speech“ were evaluated only in the assessment points t3, t4, and t5, which is concerning the fact that the child was not verbal in previous assessment points (t0, t1, and t2), and the mentioned subscales could not be assigned.

(3) Sensory profile 2 [74] is a standardized instrument for assessing children’s sensory processing patterns. It aims to identify the child’s sensory processing effects on everyday functioning at home, school, or the community. It is a caregiver/parent questionnaire consisting of 86 items. The items are estimated on a Likert-type scale (1–5). The total score on each subscale is a measure of sensory dysfunction. Higher scores point to more frequent behavior, while lower scores point to less frequent behavior. The questionnaires consist of a combination of scores, including the sensory system, behavior, and sensory pattern. They are interpreted within Winnie Dunn’s sensory profile theoretical frame (Dunn, 2014). This test was applied in all assessment points.

(4) The children’s communication checklist (CCC-2) [75] is an instrument for communication skills assessment. This instrument consists of 70 questions grouped in 10 scales: A—Speech; B—Syntax; C—Semantics; D—Coherence; E—Initiation; F—Scripted language; G—Context; H—Nonverbal communication; I—Social relations and J—Interests. The first four subscales assess different aspects of language structure, vocabulary, and discourse, and the next four scales assess pragmatic competence. The questions from the last two subscales assess behaviors we often see in children with ASD. The scale also offers the calculation of three composite scores: General Communication Composite (GCC), Social Interaction Deviance Composite (SIDC), and Pragmatic Composite (PC). However, the pragmatic composite score was used in the first version of the Communication Center for Children but is avoided in the CCC-2 version, indicating a low discriminant score value. This test was applied in t2, t3, t4, and t5 assessment points.

(5) The scale for evaluation of psychophysiological abilities of children (SEPAC) is used to evaluate whether the level of a child’s speech and language development is under the chronological age calculated in months. It involves the evaluation of receptive and expressive language development, syntax development, and proper use of prepositions, pronouns, plural, singular, case, gender, and number, as well as assessment of existing vocabulary, ability to comprehend the meaning of the spoken word, and pragmatic language skill development (ability to use language appropriately) in addition to an analysis of the child’s ability to acquire new knowledge. The SEPAC is composed of subscales specific for different years of age. By applying a subscale specific for chronological age, the level of speech and language development is estimated. These measures are regularly used in Serbia’s speech and language clinical practice [76,77,78,79,80]. The test was applied to all assessment points.

(6) EEG Recordings

The child was placed in a comfortable sitting position during the EEG recording in a soundproof and electrically shielded room. The participant was isolated from visual and auditory stimuli using white curtains arranged in a box-shaped space in a quiet room.

EEG was recorded using the Nihon Kohden Corporation, EEG 1200K Neurofax apparatus with Electrocap, International, Inc., Ag/AgCl ring electrodes filled with electro-conductive gel, providing 19 EEG channels. Electrodes were positioned according to the 10/20 placement system in the longitudinal, monopolar montage. The reference electrode was set offline to A1 and A2 (ear lobes). The horizontal and vertical electrooculograms (EOG) were recorded to detect eye blinks and eye movements. The heart rate and hand movement sensors and electrodes for jaw muscle activity were used for offline artefact removal. The AC filter was on, and the sampling rate was 200 Hz.

Impedance was below 5 kΩ. The lower filter was set to 0.53 Hz and the upper filter to 35 Hz to select frequency bands of interest and cut off higher frequencies that might be muscle artefacts. According to the International 10/20 system of electrode positioning, the following cortical regions were analyzed: Fp1-Fp2 (frontopolar), F3-F4 (mid frontal), F7-F8 (inferior frontal, anterior temporal, frontal-temporal), T3-T4 (mid temporal), T5-T6 (posterior temporal), Fz (frontal midline central), C3-C4-Cz (central), P3-P4 (parietal), Pz (parietal midline central), and O1-O2 (occipital).

In the spontaneous resting state (task 1), EEG was recorded for 2 min. The participant’s task was to keep her eyes open. The parent helped the participant minimize her movements (eye blink, head and limbs movement) as much as possible to eliminate artefacts from raw EEG traces. The second task (task 2) was a recording of 2 min EEG during listening to a simple short story (“The Red Riding Hood”). The child’s task was passive listening.

Recordings of task 1 and task 2 were used to determine the mean spectral power values for Theta (4–8 Hz), Alpha (8–12 Hz), and Beta (13–24 Hz) in all 19 electrodes.

Before analysis, we first removed the data segments, which contained obvious eye blinking, high amplitude, high-frequency muscle noise, and other irregular artefacts using ICA (EEGLAB) [81]. In our study, we used Fast Fourier Transform (FFT) to separate brain rhythms from the raw EEG trace.

The first task in the signal analysis was to choose three epochs from the recorded 2 min of the recorded signal (during task 1 and task 2, respectively): 10 s from the beginning, the middle, and the end of the recording. Before computing FFT, each epoch was multiplied by an appropriate windowing function (Hanning window) to avoid boundary leakage. Then, FFT was computed to create spectrograms and amplitude maps of the selected epoch. The mean spectral power (averaged for epochs) was further used in statistical analysis for each rhythm.

According to the established research procedure, the EEG recording was performed at all assessment points.

### 2.4. Statistical Analysis

Descriptive statistics were calculated for the achievements on different behavioral assessment scales (GARS-3, Sensory profile-2, CCC-2, and the scale for evaluation of psychophysiological abilities of children) and the achievements on cognitive assessment scales (RTČ-P, REVISK). Bivariate correlation and its statistical significance between all parameters were calculated. Statistical Package for the Social Sciences version 22.0 was used.

## 3. Results

### 3.1. Cognitive Profile

Cognitive assessments indicate stable progression and increased VIQ and PIQ values after time point t2. Quantitative indicators of achievement at different time points have been shown in Table 1.

### 3.2. ASD Symptoms

The GARS 3 results (Table 2) indicate that the girl showed very high autism index scores at the initial assessment, as well as two following assessments (t0, t1, and t2), which indicated severe ASD symptoms and a very probable presence of ASD (level 3). At points t3 and t4, autism indices decreased, and their quantitative values belonged to level 2. At point t5, autism index continued to decrease. The final testing results indicated that autistic symptoms were no longer clinically significant.

### 3.3. Sensory Profile

The results in Table 3 indicated that at the beginning (time point t0), the child’s sensory processing significantly deviated from the norm. Scores of ±1 or ±2 SD are displayed in the quadrants section, sensory section, and behavioral section. After that, we can see a gradual tendency to approach the average values at different time points. At the last time point (t5), the girl’s sensory processing was like in typically developing children on most subscales.

### 3.4. Language and Socioemotional Development

The results shown in Table 4 presented an increase in scores on both GCC and SIDC, indicating general language use, and a decrease in deviations in social interactions. Likewise, an increase in scores on the SEPAC subscale for speech and language development assessment indicates progress in receptive and expressive language.

### 3.5. EEG Findings

There were no statistically significant differences in spectral power (SP) mean value for all assessment points for a low alpha, high alpha, full alpha, and beta rhythm. There was also no statistically significant difference in spectral power of theta rhythm or pattern between time points for any electrode (Table A1). Regularity was observed only between the difference of mean values of left and right frontal spectral power theta rhythm (Table A2) during auditory-verbal information processing.

It has been observed that there is a kind of regularity only within the theta rhythm, which may be considered in analyzing auditory-verbal information processing in a child. Table 5 shows the results of theta spectral power during auditory-verbal processing.

To compare the time dynamics of specific functions measured by applied tests in the observed time interval of 2.5 years, we scaled the results of the tests to the range of 0 to 1. Specifically, we first translated the values of all observed quantities to a minimum value of 0 by subtracting from all values of each quantity the smallest value; for example, for the autism index (Table 2), we subtracted 52 from all values, so the value range 126–52 was translated into the 74–0 range. Then, the range was normalized to the 0–1 range by dividing all values of the observed value by the newly obtained maximal value. In the example for the autism index, all values were divided by 74. The obtained scaled results for all observed functions are given in Table 5.

The bivariate correlation results indicate that the observed difference of the averaged difference between the mean values of the theta rhythm spectral powers in the left and right hemispheres is highly correlated with all the examined functions (Table 6). There is also a high statistically significant correlation between all observed functions except between sensory profile and PIQ (*p* = 0.098).

Although this correlation is high, it can be noticed that the change in all functions except EEG is small in the critical period of the first year of treatment (Figure 1). Namely, Table 5 shows that the absolute change between the periods t0 and t1 for sensory profiles, GARS, speech–language development, VIQ, and PIQ is in the range of 0.01 to 0.043. If, however, one looks at the change in the normalized value of the theta rhythms Fnorm, that change is 0.199.

## 4. Discussion

The paper presents the results of a two-year treatment follow-up study of a child with overlapping autism spectrum disorder and specific language impairment symptoms between 36 and 66 months of age. To evaluate the dynamics of therapy progress, we obtained results about the cognitive profile, autism risk index, sensory and speech–language profile, and EEG findings in six assessment points.

After the first six months of treatment, it was noticed that there were minimal changes in the retest findings obtained by behavioral tests. These findings were in line with the clinical observation of the child (there was no significant progress in the child’s developmental abilities despite intensive integrative therapy). In the first assessment point t1 (after six months of treatment), the only noticeable difference was noticed in EEG findings. Hence, question arose about whether the estimation of EEG correlates of auditory-verbal processing may predict the treatment efficacy when behavioral tests do not register examined functions changes.

### 4.1. Cognitive Profile

Analysis of the results related to cognitive profile at all six assessment points clearly shows that the values of both verbal IQ and performance IQ have been continuously rising from the assessment point t2. Findings in the literature related to the cognitive profile of children with ASD go in two directions. While some propose that ASD is linked with low intelligence [82], others documented that ASD is genetically linked with high intelligence [83,84,85]. Considering that at the t3 assessment point, the child`s performance IQ was 94, and it was stable at the t4 and t5 assessment points (Table 1), this is in line with the description of the cognitive profile of children with SLI, which is characterized by a normal nonverbal cognitive function in which performance IQ is greater than 85 [2].

### 4.2. Autism Risk Score

The severity of ASD symptoms, measured by GARS, decreased from the t2 assessment point. A more noticeable decrease happened in the periods t3 and t4 (at the age of 54 and 60 months), while at 66 months, the autism index score indicated no autism (Table 2). Considering the literature describing children with ASD who have repetitive and stereotypical behavior compared with SLI children [86,87], the examined child exhibited the same forms of behavior, especially during the first year of treatment. On the other hand, 90% and more children with ASD have sensory abnormalities [88]. These children showed abnormalities in auditory discrimination [89], difficulties in visual abilities [90], tactile hypersensitivity [91], and food selectivity as well [92]. The decrease in these autistic symptoms noted in our study is in relation to the findings that proved that some children changed category from severely to mildly autistic or no longer autistic due to the intervention program. According to screening methods, it was concluded that individually tailored psycho-educational therapy had a significant effect on autism severity [93]. Our findings are in accordance with the presented data, as the girl from our study changed categories from high autism probability to no autism probability.

### 4.3. Sensory Profile

The sensory profile specifically changed as therapy passed, similarly to the severity of autism symptoms. Similar to the first assessment point, with results presented in Table 3 (hyper reaction profile, except for visual sensitivity), in the second assessment point and even in the third assessment point, the girl showed severely and significantly weakened sensory capacities, with lower sensory adaptation and more generalized and more serious deficits in all other subdomains, which is similar with the previous study by Panerai et al. [94]. After that, sensory capacities stabilize at points t3 to t6. These results can favor sensory integration therapy and re-education of psychomotor skills. Even though the authors argue that sensory abnormalities are difficult to treat, our study is in accordance with other research, which indicated that children who had targeted interventions show significantly greater improvement in sensory processing [95].

### 4.4. Speech–Language Profile

At the first assessment point (t0), the findings on the SEPAC showed that the child was non-verbal, without the use of functional words, indicating that the child fell significantly behind the chronological age considering speech–language development. During the later assessment periods (t1, t2, t3, t4, and t5), especially from the t2 period or 48 months of the child’s age, the girl showed progress in speech-language development. Accordingly, from the t2 assessment point, the girl was tested not only by the SEPAC but also with the children’s communication checklist (CCC-2), which is an instrument for the assessment of communication skills.

However, the speech–language profile has changed significantly since the t3 assessment point, or 54 months of the child’s age (Table 4). In that time, the girl increased scores on both general communication and social interaction deviance composite, indicating progress in language use in general and a reduction in deviations in social interactions. Although the estimation of speech and language development by the SEPAC subscale showed progress in receptive and expressive language at 48 months, it was especially noticed at the 54, 60, and 66 months (Table 4). At 54 months, the child was verbal using functional agrammatical speech. She expressed her needs verbally and communicated with people from the environment. Her speech–language ability was further improved at 60 and 66 months, although reduced in particular segments of grammar and vocabulary. Studying communicative profiles in children with ASD, developmental delays, and typical development [96], the authors pointed to deficits in communication rate and joint attention as the strongest predictors of communicative skills at age 3 in children with ASD. Additionally, in our case report, comparing linear approximation of speech–language development and the difference in activation in the left and right hemispheres in EEG findings, it has been noticed that these two functions coincide. This coincidence is maximally reflected in the last two time points (60 and 66 months), which are almost identical, indicating a strong connection between speech–language development and EEG correlates (theta rhythm).

It is consistent with findings indicating that increased frontal and temporal theta is a predictor of better language and cognitive skills [41]. Furthermore, other authors documented the role of frontal theta in future cognitive abilities in children [39].

### 4.5. EEG Findings

Analysis of EEG correlates during auditory stimulation for all electrodes revealed no regularity of timer pattern in any spectral band (alpha, beta, theta). We obtained some regularity when considering the difference between the left and right frontal regions in theta rhythm (Table A2). In the first three assessment points (36, 42, 48 months), frontal theta rhythm has been dominant over the right hemisphere. After that, in the other three assessments points (54, 60, 66 months), frontal theta has been lateralized on the left hemisphere.

Our findings are similar to some other neurophysiological results. During speech perception, theta waves are prominent in younger children [97], specifically in the anterior brain regions in the first four years of healthy children [47,49]. In right-handed children left hemisphere is dominant during speech perception [98].

Some fMRI studies compared language lateralization during auditory tasks between youngest children and adults, while studies reported activation only in the left hemisphere [99,100]. In contrast, the left lateralization increases with age [101]. Other authors pointed out that the youngest children have left but equally have right hemisphere activations [102], specifically in the inferior frontal gyrus and superior temporal gyrus. At older ages, activation of the right hemisphere decreases [103]. These findings are similar to our EEG results showing the alternation of theta rhythm dominance from the right to left hemispheres at 54 months.

On the other hand, the noticeable difference in EEG findings assessed in the first assessment point (even though other behavioral tests do not register changes in the level of examined functions) may point to treatment efficacy in children with ASD and SLI.

### 4.6. Correlations

Summarizing and analyzing all obtained results showed a statistically significant correlation between all observed functions except sensory profile, and PIQ (Table 6) with EEG correlates related to the difference between the average mean values of theta rhythm in the left (F1, F3, F7) and right (F2, F4, F8) frontal region (Table A2). Of the statistically significant correlations, the lowest correlation is between sensory profile and EEG correlates (Pearson coefficient −0.818), while the strongest correlation is between GARS and VIQ (Pearson coefficient −0.999). It can be noticed that some correlations are positive and some negative. For example, GARS is positively correlated with sensory profile and negatively correlated with all other functions.

It can be noticed that a higher level of speech and language abilities, VIQ, PIQ, and EEG correlates are in relation to lower GARS symptoms. As some authors mention, verbal abilities play an important role in ASD symptom changes [104]. Additionally, it has been shown that higher verbal intelligence predicted lower social disability and higher communication ability. In contrast, higher nonverbal intelligence predicted higher communication ability [105], which indicates that verbal competencies improve social communication, consequently leading to a reduction in ASD symptoms. Furthermore, some findings indicate a relationship between ASD and EEG abnormalities. The management of seizure waves in children diagnosed with ASD may improve function and reduce autistic symptoms [106].

### 4.7. Time Dynamics

The temporal dynamics of the examined processes point out that some of them have positive (Speech and Language, PIQ, VIQ, and EEG) and some negative (GARS and sensory profile) trends (Figure 1). Those findings are not contradictory, and they are the consequence of scoring. Higher values on GARS indicate a higher level of autism spectrum disorder, while higher values on VIQ reflect a higher level of verbal intelligence.

Figure 1 shows that the behavioral tests after six months had a minor change compared to the start of treatment (on the *x*-axis difference between points t0 and t1). Only a change in EEG values is observed. While the change in all functions measured by behavioral tests is small (<4%), the change in the observed difference between averaged mean values of theta rhythms of the left and right hemispheres is about 20% (19.9)

Although it could be speculated that this change is conditioned by maturation, it is difficult to assume that only one function changes so much on that basis, while the change in other functions is minimal. Our claim is supported by the fact that there is a high correlation between the examined functions in the whole observed period. Therefore, it seems that this change is much more likely to be a consequence of treatment. Furthermore, it indicates a period of accumulation necessary for specific accumulated knowledge to manifest itself, which is in relation to Luria’s theory of neuropsychological rehabilitation [107]. Accordingly, given the course of development of all developmental functions, the level of this change after six months can be related to progress due to the applied treatment and not to natural maturation processes. This may also be supported by literature findings indicating that early intervention provides the best opportunity for optimal child development [57,108]. Studies show that early treatment can improve brain functioning in children with ASD. Early behavioral treatment with preschool children with ASD led to activations in the left frontal and right temporal regions in two children [109] and in five children [110]. One recent EEG study showed greater mu rhythms in 18–30-month-old toddlers diagnosed with ASD who received ESDM treatment, compared with ASD children without treatment, but in social skills such as recognizing familiar persons [111]. Likewise, greater frontal theta in 12-to-18-month-old toddlers were observed during the early intervention [112].

It is also important to point out that even though the child at the beginning of the treatment had overlapping ASD and SLI symptoms, she changed the category from severely to mildly autistic, or no longer autistic at the age of 66 months when the autism index score indicated no autism. Additionally, the progress in receptive and expressive language was noticed at the age of 48 months. Based on the obtained results, it may be considered that the child has progressed in terms of psychophysiological development due to intervention and treatment efficacy.

An important note is that the EEG correlates predicted the progress in therapy after the six months of treatment. At this time point, this information was essential because it provided information on whether the treatment is effective and whether changes are needed in its organization and planning. Accordingly, the estimation of EEG correlates during auditory-verbal information processing in our case report was diagnostic support for assessing treatment efficacy.

In general, such findings, which are consistent with our study findings, suggest the importance of early intervention and the role of EEG neural biomarkers early in life that can lead to neural brain reorganization and the absence of ASD symptoms [113]. The results obtained in the case report may be of significant clinical value in cases where behavioral tests do not indicate treatment progress. This is especially important in the treatment of young children when time is a critical factor [66] because neurophysiological processes are highly dynamic, and time desynchronization can have significant consequences on the course of treatment. Further research on a larger population of children will provide a better understanding of the interconnectedness in scientific terms.

## 5. Conclusions

In cases when behavioral tests do not show improvement in a child’s abilities and the treatment lasts a long time, the question of the treatment efficacy arises. The application of a non-invasive EEG technique for monitoring cognitive and speech–language information processing may be an objective prediction method of rehabilitation achievement in children with overlapping ASD and SLI symptoms. More precisely, the estimation of EEG correlates of auditory-verbal stimulation may predict child progress in terms of applied therapy in conditions when speech–language and behavioral tests do not register changes in the level of examined functions. The positive shift in EEG findings after the six months of integrative therapy, and the lack of response to behavioral tests, can be explained by the cumulative effect of intensive therapy, which manifests itself later in behavioral functions. In such a way, it may impact the organization and planning of further treatment of children with SLI and ASD. In addition to this essential finding, this case report points to the subsequent progress of the child during the two and a half years of integrative therapy in terms of all observed functions, including speech and language, showing that the examined child developed her speech–language ability and reduced autistic symptoms to the level of no autism probability. Such results indicate the importance of early and continuous integrative therapy.

## Figures and Tables

**Figure 1 diagnostics-12-01110-f001:**
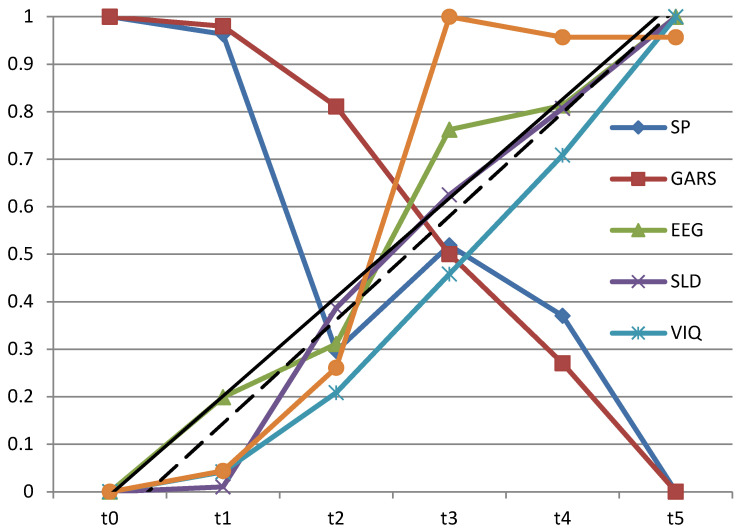
Comparison of the normalized achievement in all evaluation tests in all assessment points. *Note*: SP—sensory profile; GARS—Gilliam Autism Rating Scale; EEG—electroencephalography; SLD—speech–language development; VIQ—verbal IQ; PIQ—performance IQ.

**Table 1 diagnostics-12-01110-t001:** Cognitive assessment at different time points.

Time	t0	t1	t2	t3	t4	t5
Age (in years)	3;0	3;6	4;0	4;6	5;0	5;6
VIQ	50	51	55	61	67	74
PIQ	71	72	77	94	93	93

*Note:* VIQ-verbal IQ; PIQ-performance IQ.

**Table 2 diagnostics-12-01110-t002:** Results of the GARS-3 in all assessment points.

Time	t0	t1	t2	t3	t4	t5
**Age (in months)**	36	42	48	54	60	66
**Restricted/repetitive behaviors**	15	14	12	7	5	4
**Social interaction**	14	13	14	9	7	3
**Social communication**	12	12	12	8	7	5
**Emotional responses**	14	14	9	7	5	3
**Cognitive style**	NA	NA	NA	11	9	7
**Maladaptive speech**	NA	NA	NA	10	8	5
**Sum of the standard scores**	55	53	47	52	41	27
**Autism Index**	126	123	112	89	72	52
**Degree of difficulty**	level 3	level 3	level 3	level 2	level 2	/

*Note*: NA—not assigned (subscales were not assigned because the child did not speak during t0, t1, and t2).

**Table 3 diagnostics-12-01110-t003:** Results of the sensory profile 2 in all assessment points.

Sensory Profile		Time and Age (in Months)											
		t0		t1		t2		t3		t4		t5	
		36		42		48		54		60		66	
		Score	SD	Score	SD	Score	SD	Score	SD	Score	SD	Score	SD
Quadrants	Seeking	38	X¯	35	X¯	17	−1	20	X¯	28	X¯	23	X¯
	Avoiding	78	+2	75	+2	57	+1	58	+1	47	+1	35	X¯
	Sensitivity	73	+2	78	+2	51	+1	55	+2	50	+1	50	+1
	Registration	63	+2	52	+1	40	X¯	40	X¯	32	X¯	30	X¯
Sensory section	Auditory	36	+2	36	+2	26	+1	28	+1	28	+1	11	X¯
	Visual	6	−1	7	−1	12	/	10	X¯	8	−1	10	X¯
	Tactile	21	X¯	23	+1	5	−1	20	X¯	17	X¯	9	X¯
	Body position	16	+1	13	X¯	8	X¯	7	X¯	8	X¯	8	X¯
	Movement	10	X¯	10	X¯	5	−1	7	X¯	10	X¯	8	X¯
	Oral	46	+2	40	+2	34	+2	40	+2	21	X¯	20	X¯
Behavioral section	Conduct	29	+1	32	+2	19	X¯	34	+2	28	+1	19	X¯
	Social Emotional	62	+2	65	+2	48	+2	45	+2	43	+2	35	X¯
	Attentional	30	+1	28	+1	20	X¯	18	X¯	28	+1	20	X¯

*Note:* SD—Standard deviation from the mean population achievement. +1SD means more than others, −1SD means less than others. +2SD means much more than others, −2SD means much less than others. X¯-average-score indicates that sensory processing is just like the majority of others.

**Table 4 diagnostics-12-01110-t004:** Results of the children’s communication checklist for four assessment points (CCC-2) and the scale for evaluation of psychophysiological abilities of children for six assessment points (SEPAC).

Time	t0	t1	t2	t3	t4	t5
**Age** **(in months)**	36	42	48	54	60	66
**CCC-2: GCC**	NA	NA	2	17	28	43
**CCC-2: SIDC**	NA	NA	2	16	35	41

**SEPAC:ESLD**	9	13	22	30	41	51

*Note:* NA—not assigned; CCC-2 could not be administered before the age of 4; GCC—General Communication Composite; SIDC—Social Interaction Deviance Composite; ESLD—estimated age based on the level of speech and language development.

**Table 5 diagnostics-12-01110-t005:** Normalized values of sensory profile, GARS, EEG findings, speech–language profile, VIQ, and PIQ in the range of 0 to 1.

	Sensory Profile	GARS	EEG Findings	Speech–Language Profile	VIQ	PIQ
t0	1	1	0	0	0	0
t1	0.963	0.98	0.199	0.01	0.042	0.043
t2	0.296	0.811	0.311	0.386	0.208	0.261
t3	0.518	0.5	0.762	0.625	0.458	1
t4	0.370	0.270	0.813	0.807	0.708	0.956
t5	0	0	1	1	1	0.956

**Table 6 diagnostics-12-01110-t006:** Pearson correlation and obtained statistical significance between normalized test results.

Correlations							
		Sensory Profile	GARS	EEG Findings	Speech-Language Profile	VIQ	PIQ
**Sensory profile**	Pearson Correlation	1	0.846 *	−0.818 *	−0.907 *	−0.858 *	−0.732
	Sig. (2-tailed)		0.034	0.047	0.013	0.029	0.098
	N	6	6	6	6	6	6
**GARS**	Pearson Correlation	0.846 *	1	−0.968 **	−0.979 **	−0.999 **	−0.910 *
	Sig. (2-tailed)	0.034		0.002	0.001	0.000	0.012
	N	6	6	6	6	6	6
**EEG**	Pearson Correlation	−0.818 *	−0.968 **	1	0.971 **	0.959 **	0.966 **
	Sig. (2-tailed)	0.047	0.002		0.001	0.002	0.002
	N	6	6	6	6	6	6
**Speech-language profile**	Pearson Correlation	−0.907 *	−0.979 **	0.971 **	1	0.975 **	0.934 **
	Sig. (2-tailed)	0.013	0.001	0.001		0.001	0.006
	N	6	6	6	6	6	6
**VIQ**	Pearson Correlation	−0.858 *	−0.999 **	0.959 **	0.975 **	1	0.889 *
	Sig. (2-tailed)	0.029	0.000	0.002	0.001		0.018
	N	6	6	6	6	6	6
**PIQ**	Pearson Correlation	−0.732	−0.910 *	0.966 **	0.934 **	0.889 *	1
	Sig. (2-tailed)	0.098	0.012	0.002	0.006	0.018	
	N	6	6	6	6	6	6

*. Correlation is significant at the 0.05 level (2-tailed). **. Correlation is significant at the 0.01 level (2-tailed).

## Data Availability

Not applicable.

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
