# Peer review of "Can EEG Correlates Predict Treatment Efficacy in Children with Overlapping ASD and SLI Symptoms: A Case Report"

_diagnostics, 2022, doi:10.3390/diagnostics12051110_

Round 1

Reviewer 1 Report

This paper by Maksimovic et al describes the results of a  prospective, longitudinal study of a child with overlapping autism spectrum disorders (ASD) and specific language impairment (SLI) symptoms.

In the two and a half years of continuous integrative therapy, behavioral tests and EEG findings were evaluated to estimate which variable best describes treatment progress. At the end of the treatment they find that improvements of EEG correlates could be indicative of a good response to rehabilitation treatment.

1) According to DSM 5 (American Psychiatric Association (APA). (2013). Diagnostic and Statistical Manual of Mental Disorders, 5th edition (DSM-5).Washington, DC: American Psychiatric Publishing.) language impairment/communication disorders is inherent in ADS, so it is necessary to explain better what level of speech impairment the patient presented and how it was different from the typical impairment in ASD.

2)Line 159: the child was non-verbal. How can evaluate in this case a specific language impairment?

3) Line 259:How was it possible to apply verbal subtests to the child with non-verbal form of autism?

Table 2: "degree of dificulty" change with "degree of difficulty"

Author Response

Title: Can EEG correlates predict treatment efficacy in children with overlapping ASD and SLI symptoms: A case report

Diagnostics

Section: Pathology and Molecular Diagnostics

Special Issue: "Advances in the Diagnosis and Management of ENT Diseases"

Dear Reviewer,

Please find enclosed the Review Report of our manuscript entitled "Can EEG correlates predict treatment efficacy in children with overlapping ASD and SLI symptoms: A case report".

Thank you for acknowledging the manuscript topic related to the estimation of EEG correlates in a child with overlapping autism spectrum disorders (ASD) and specific language impairment (SLI) symptoms, which can be indicative of child progress in terms of applied integrative therapy, or in other words improvements of EEG correlates could be indicative of a good response to rehabilitation treatment. Thank you for your opinion that the manuscript can be improved in the section Introduction and research design, while it must be improved in sections Methods, Results, and Conclusions. Accordingly, we have refined the manuscript according to your suggestions.

We are very grateful for your extremely valuable comments: we have modified and refined the paper according to your suggestions.

We sincerely hope that the revised version of the manuscript would receive a positive further review and that it would be accepted for the publication in the Diagnostics.

Sincerely,

Ljiljana Jeličić, Ph.D.

Ph.D. in Neurosciences, Senior Research Associate,

Speech and language pathologist,

R&D Institute "Life Activities Advancement Center"

Institute for experimental phonetics and speech pathology

  1. Jovanova No 35

11000 Belgrade, Serbia

Office phone: +381 11 3208 519

Mob phone: + 381 64 22 14 295

Email: lj.jelicic@add-for-life.com

           lilijen @ymail.com

RESPONSE FOR REVIEWER 1:

Introduction

  1. - - According to DSM 5 (American Psychiatric Association (APA). (2013). Diagnostic and Statistical Manual of Mental Disorders, 5th edition (DSM-5). Washington, DC: American Psychiatric Publishing.) language impairment/communication disorders is inherent in ADS, so it is necessary to explain better what level of speech impairment the patient presented and how it was different from the typical impairment in ASD.

Thank you very much for the well pointed comment. We apologize for the omission to more precisely describe speech impairment in our case report. I would like to explain the starting points from which we started in setting the research aim of our study. In the conditions of early diagnosis, during the first admission of a child, it is difficult to specify which diagnosis is final, especially if we take into account the fact that the child was not included in integrative therapy before the first admission, and the fact that there is often comorbidity between specific diagnoses. Speech-language deficits are characteristics of ASD and SLI, while the existence of a common profile of them is a matter of tireless debate in the research community. These findings have been mentioned and cited in the section Introduction of our paper (References 7-11, Line 46-48). These references consider the question about the link between speech-language abilities in ASD and SLI.

Considering the child's status at the first admission, the admission team set the working diagnosis: the child with overlapping ASD and SLI symptoms, with the aim that the child starts intensive integrative therapy immediately, taking into account the importance of early stimulation, which we also explained in the section Introduction (References 54-66, Line 95-111) and Discussion (Reference 93, discussion in Line 516-520). Subsequent progress of the child during the two and a half years of integrative therapy in terms of all observed functions, including speech and language, showed that the examined child developed her speech-language ability and reduced autistic symptoms to the level of no autism probability. This was noted in section Conclusion (Line 675-680).

Concerning your suggestions and assessment, we have modified and refined the paper.

Accordingly, in the section Introduction we gave an explanation what the research aim was and how it was postulated. In the section Materials and Methods, and Discussion, we gave a more detailed data on the level of a speech impairment that patient presented through all assessment points, showing her progress in speech-language development and reducing autistic symptoms. In the section Conclusions we defined the child's progress in all examine abilities. We showed that after the first six months of integrative therapy (which is a valuable period for estimation of treatment efficacy), only the examination of EEG correlates indicated progress, which was confirmed with other tests in later assessment points. In that sense, we tried better to explain in section Materials and Methods and Discussion what level of speech impairment the patient presented. It is important to emphasize that considering speech impairment, it was hard to distinguish whether the speech and language deficit manifested itself as an independent disorder or was within the ASD during the assessment at the child's first admission, which is mentioned in the description of the case report. The results were improved in clarity of presentation and explanation, and spelling errors were corrected. The conclusions were refined by an essential item that indicates the child's progress in speech-language and other observed skills due to intensive integrative therapy.

Please see Introduction, the following text in the Introduction (Line 112-122):

“It is usually not easy to specify the final diagnosis during the early diagnostic process in young children (i.e. the first admission of a child). This is especially so if the child has not previously been involved in integrative therapy, but also due to the comorbidity between certain diagnoses [7-11]. Speech-language deficits are characteristics of ASD and SLI, while the existence of a common profile of them is a matter of tireless debate in the research community [7]. In that sense, the goal of this study was not to determine the final diagnosis at the time of child admission but to monitor the child's progress due to the implementation of early integrative therapy that ensures the development of the child's existing abilities. In addition to the importance of implementing early treatment, an important aspect is the assessment of the treatment efficacy in a certain period. This aspect is essential due to the critical period of speech-language development [64-66].”

Please see Materials and Methods, the following text in the Materials and Methods (Line 174-175; 207-208; 210-219):

“At the moment of admission, the child was non-verbal (without functional words), uncooperative, and manifesting ASD symptoms.” – Line 174-175

“The girl was non-verbal. She did not produce any functional words or holophrases during the first assessment.” – Line 207-208

“Generally considered, the girl had severe receptive and expressive language deficits, and failed significantly behind the chronological age considering speech-language development. It was hard to distinguish whether the speech and language deficit manifested itself as an independent disorder or was within the ASD during the assessment at the child's first admission.

Although the child was non-verbal at the admission, it is important to emphasize that speech and language skills have developed with specific dynamics during the integrative therapy. However, it was estimated in later assessment periods when the child showed progress in speech-language development. In parallel with the development of speech and language, the girl reduced autistic symptoms as well.” – Line 210-219

Please see Discussion, the following text in the Discussion (Line 536-543; 547-554):

“At the first assessment point (t0), the findings on the SEPAC showed that the child was non-verbal, without the use of functional words, indicating that the child failed significantly behind the chronological age considering speech-language development. During the later assessment periods (t1, t2, t3, t4, and t5), especially from the t2 period or 48 months of the child's age, the girl showed progress in speech-language development. Accordingly, from the t2 assessment point the girl was tested not only by the SEPAC but also with the Children's communication checklist (CCC-2), which is an instrument for the assessment of communication skills.”- Line 536-543

“Although the estimation of speech and language development by the SEPAC subscale also indicated showed the progress in receptive and expressive language at 48 months, it was especially noticed at the 54, 60, and 66 months (Table 4). At 54 months, the child was verbal using functional agrammatical speech. She expressed her needs verbally and communicated with people from the environment. Her speech-language was further improved at the 60 and 66 months, although reduced in particular segments of grammar and vocabulary.” - Line 547-554

Please see Conclusions, the following text in the Conclusions (Line 675-680):

“Besides this essential finding, this case report point to the subsequent progress of the child during the two and a half years of integrative therapy in terms of all observed functions, including speech and language, showing that the examined child developed her speech-language ability and reduced autistic symptoms to the level of no autism probability. Such results indicate the importance of early and continuous integrative therapy.” - Line 675-680

Materials and Methods

  1. - Line 159: the child was non-verbal. How can evaluate in this case a specific language impairment?

Thank you for your observation. Before stating what we have elaborated in the paper in relation to this issue, we would like to explain why we have made such a claim. So, at the first assessment point, the Scale for evaluation of psychophysiological abilities of children (SEPAC) was used to evaluate whether the level of a child's speech and language development is under the chronological age calculated in months. This scale is described in the section Materials and Methods (Line 338-349).  At the first assessment point (t0), the findings on the Scale showed that the child was non-verbal, without the use of functional words, indicating that the child failed significantly behind the chronological age considering speech-language development. During the later assessment periods (t1, t2, t3, t4 and t5), and especially from the t2 period, the girl showed progress in speech-language development. To be more precisely, from the t2 assessment point the girl was tested not only by the SEPAC, but also with the Children's communication checklist (CCC-2), which is an instrument for the assessment of communication skills (Line 326-337). Accordingly, it is noted in the section Method and Materials (Study design) that the dynamics of the child’s progress conditioned the application of mentioned tests at later ages (Line 263-264).

According to your question, we have also introduced the following description in the Methods section:

Please see Materials and Methods, the following text in the Materials and Methods (Line 210-219):

“. Generally considered, the girl had severe receptive and expressive language deficits, and failed significantly behind the chronological age considering speech-language development. It was hard to distinguish whether the speech and language deficit manifested itself as an independent disorder or was within the ASD during the assessment at the child's first admission.

Although the child was non-verbal at the admission, it is important to emphasize that speech and language skills have developed with specific dynamics during the integrative therapy. However, it was estimated in later assessment periods when the child showed progress in speech-language development. In parallel with the development of speech and language, the girl reduced autistic symptoms as well.”

  1. Line 259: How was it possible to apply verbal subtests to the child with non-verbal form of autism?

Thank you for your observation. We apologize if we did not precisely describe when the verbal subtests were applied. Starting from Line 285 we described REVISK (revised Wechsler Intelligence Scale for Children), stating that this instrument provides insight into total, verbal and performance scores. This test was applied when the girl was 5 years old, or more precisely in t4 and t5 assessment points. At that assessment point, the girl was verbal and she has developed speech-language abilities.

.

According to your suggestion, we tried to be more precise in defining this, and in that sense, we added this description related to the girl's age and verbal status (Line 291-292).

Please see Materials and Methods, the following text in the Materials and Methods (Line 293-294):

“At these two assessment points (60 and 66 months of age), the girl was verbal and had developed speech-language abilities.” 

Results

  1. - Table 2: "degree of dificulty" change with "degree of difficulty"

Thank you for your suggestion. We sincerely apologize for this omission. We have corrected the observed misspellings mistake in the Table 2.

Please see Table 2 (last row in the Table 2).

According to your suggestion, we have also made the proposed moderate changes in the English language in the paper.

Reviewer 2 Report

This is a case report on a follow up of a girl with autism spectrum disorder and language and specific language impairment. The case report is very interesting; there are only misspellings that should be corrected, for example 

Line 153 pre-

table 2: restricted (not eestricted) 

Author Response

Title: Can EEG correlates predict treatment efficacy in children with overlapping ASD and SLI symptoms: A case report

Diagnostics

Section: Pathology and Molecular Diagnostics

Special Issue: "Advances in the Diagnosis and Management of ENT Diseases"

Dear Reviewer,

Please find enclosed the Review Report of our manuscript entitled "Can EEG correlates predict treatment efficacy in children with overlapping ASD and SLI symptoms: A case report".

Thank you for acknowledging our study and its relevant topic that is referred to a case report on a follow-up of a girl with autism spectrum disorder and specific language impairment. Thank you for your opinion that the manuscript is well written with an adequate introduction, methods, clearly presented results, and conclusions.

We are very grateful for your suggestions considering the English language and the necessity to correct misspellings. We sincerely apologize for these mistakes. Accordingly, we have conducted fine/minor spell checking, but also moderate changes in the English language, which has resulted in an improved version of our manuscript.

We hope that the revised version of the manuscript would receive a positive further review and that it would be accepted for the publication in the Diagnostics.

Sincerely,

Ljiljana Jeličić, Ph.D.

Ph.D. in Neurosciences, Senior Research Associate,

Speech and language pathologist,

R&D Institute "Life Activities Advancement Center"

Institute for experimental phonetics and speech pathology

  1. Jovanova No 35

11000 Belgrade, Serbia

Office phone: +381 11 3208 519

Mob phone: + 381 64 22 14 295

Email: lj.jelicic@add-for-life.com

           lilijen @ymail.com

RESPONSE FOR REVIEWER 2:

  1. Materials and methods

Line 153 pre-

Thank you very much for your observation. We sincerely apologize for this omission and other misspelling mistakes. We corrected the proposed misspellings.

Please see Materials and Methods, the following text in the Materials and Methods (Line 168):

“Pre-lingual speech phases occurred regularly, but the lingual phase was inconsistent after the babbling phase.”

  1. Results

table 2: restricted (not eestricted)

Thank you very much for your observation. We sincerely apologize for this omission. According to your observation we have corrected the mentioned spelling mistake.

Please see Table 2 (the third row in the Table 2).

We would like to mention that we have also checked the English language and style in the manuscript, as you proposed to us in the review (fine/minor spell check required). Besides, we have also conducted moderate changes in the English language.  We sincerely apologize again for such mistakes.

Round 2

Reviewer 1 Report

I really appreciate the changes made, now I think it is clearer and more acceptable for the final version.